# Evaluation of the Protective Effect of Vitamin B17 Against the Potential UV Damage Using *Drosophila* as a Model

**DOI:** 10.3390/insects16121238

**Published:** 2025-12-08

**Authors:** Hanaa Elbrense, Mohamed T. Yassin, Karim Samy El-Said, Ahmed Said Atlam, Samar El-Kholy

**Affiliations:** 1Department of Zoology, Faculty of Science, Tanta University, Tanta 31527, Egypt; hanaa.elbrense@science.tanta.edu.eg (H.E.);; 2Division of Biochemistry, Department of Chemistry, Faculty of Science, Tanta University, Tanta 31527, Egypt; 3Department of Physics, Faculty of Science, Tanta University, Tanta 31527, Egypt

**Keywords:** Amygdalin, *Drosophila*, ultraviolet radiation, oxidative stress, cellular alterations

## Abstract

Commercial promotion claims that vitamin B17 has a protective role against cancer. The current study, using a UV-exposed *Drosophila* model, confirmed that B17 has a protective role at the physiological level and to some extent can improve symptoms at the tissue and cellular levels.

## 1. Introduction

Dietary supplements are now a common part of daily life for many people, ranging from athletes looking to boost performance to individuals striving to meet specific nutritional requirements [1]. Their widespread use raises critical questions about potential benefits and risks. Are these supplements vital, or are they simply a product of clever marketing? This provides a foundation for conducting independent research into the potential benefits and risks of each type.

Among the most controversial nutritional supplements is amygdalin, also known as vitamin B17. It is a cyanogenic glycoside composed of a sugar moiety (two glucose units) linked to mandelonitrile, a benzaldehyde-cyanide derivative. It is a naturally occurring compound found in the seeds of some fruits, such as apricots, peaches, and apples [2]. Despite not being a vitamin, meaning it is not essential to maintain human health, survival and growth, it has been labeled as a vitamin. This encourages many people to take it as part of metabolic therapy programs despite significant concerns about its misuse [3]. Upon enzymatic hydrolysis, B17 can release glucose, benzaldehyde, and small amounts of cyanide, which can interact with cellular redox systems. These chemical products are believed to modulate reactive oxygen species (ROS) levels, enhance antioxidant defenses, stabilize mitochondrial membranes, and regulate apoptosis. Through these biochemical actions, B17 may protect cells from oxidative stress and environmental insults such as ultraviolet radiation (UVR). These low-level stress responses are consistent with the concept of hormesis, where subtoxic doses of a compound elicit beneficial cellular effects. This mechanism highlights the need for careful interpretation given the compound’s controversial status [4].

Ultraviolet radiation accounts for approximately 7–9% of the solar energy that reaches Earth’s surface [5]. UVR spans wavelengths from 100 to 400 nanometers and is categorized into three bands based on wavelength: ultraviolet A (UVA; 315–400 nm), ultraviolet B (UVB; 280–315 nm), and ultraviolet C (UVC; 100–280 nm) [6]. UVR is considered one of the most common environmental causes of cancer, particularly skin cancers. Prolonged exposure to UVR from the sun or artificial sources like tanning beds can damage the DNA in skin cells, leading to mutations that promote cancer development, disrupt antioxidant defense systems, increase reactive oxygen species (ROS) [7], leading to lipid and protein oxidation and DNA damage, thereby accelerating processes such as photocarcinogenesis and aging [8,9,10].

Based on the known antioxidant and cytoprotective properties of vitamin B17, we hypothesize that B17 supplementation can confer resistance to UVR in *Drosophila melanogaster* (Diptera: Drosophilidae) by reducing oxidative stress, preserving mitochondrial and tissue integrity, and limiting apoptosis. To test this hypothesis, the current study investigates the potential protective role of B17 at multiple levels—physiological, biochemical, and ultrastructural—focusing on mortality, reproductive performance, and the compound eye, which is particularly susceptible to UVR-induced damage [11]. *D. melanogaster* offers a reliable and convenient system for studying cellular responses to oxidative stress and environmental insults [12], providing an ideal model to assess multi-level endpoints. Many fundamental cellular and molecular pathways (e.g., DNA repair, oxidative stress response, apoptosis) are conserved between *Drosophila* and higher organisms, including humans [13,14]. By integrating these measures, our study aims to provide comprehensive evidence for the cytoprotective and therapeutic potential of B17 against UVR-induced cellular and organismal damage.

## 2. Material and Methods

### 2.1. Drosophila Strain and Rearing

In this study, wild-type *D. melanogaster* (Canton S #64349), sourced from the Bloomington *Drosophila* Stock Center at Indiana University, Bloomington, IN, USA, was used. Flies were maintained on standard corn meal-agar media under controlled laboratory conditions of 25 ± 2 °C, 60–80% relative humidity, and a 12:12 h light/dark cycle [15]. The rearing and handling of *Drosophila* complied with the ethical standards set by the Research Ethics Committee at the Faculty of Science, Tanta University (Institutional Animal Care and Use Committee SCI-TU-0476).

### 2.2. Effect of Ultraviolet Radiation Exposure

In this study, UVR exposure was conducted using a single 60 cm fluorescent ultraviolet lamp rated at 100 watts as previously described [16]. The lamp was housed in a wooden box with internal dimensions of 60 × 60 × 60 cm^3^ (Figure 1). To enhance light reflection and minimize dispersion, the box’s interior surfaces were coated with a reflective silver coating.

The test *Drosophila* were positioned at a fixed distance of 30 cm from the UV lamp during exposure. The spectral output of the light was analyzed using an Ocean Optics RED TIDE USB650 UV spectrometer (Ocean Optics, Inc., Duiven, The Netherlands). The emission spectrum showed prominent peaks at 365 nm (100% relative intensity) and 333 nm (8% relative intensity), both within the UV-A range (315–400 nm). Additionally, a more energetic peak at 313 nm, with a relative intensity of 27%, was detected, within the UV-B range (280–315 nm) Thus, the adult *Drosopila* were exposed to UVR at a dosage of 54,000 J/m^2^ for one hour daily for a week.

### 2.3. Vitamin B17 Treatment Protocol

Vitamin B17 (Amygdalin/Laetrile) was obtained from Al Nahdi Pharmacy, KSA (Jeddah, Saudi Arabia), with a reported purity of 100%. B17 was incorporated into the standard *Drosophila* diet at a concentration of 30 µg/L distilled water in accordance with [17]. This value denotes the concentration to which adult *Drosophila* were exposed during the experiment, and should not be interpreted as the actual amounts ingested or absorbed. The compound was thoroughly mixed into the fly medium to ensure homogeneous distribution. Flies in the treatment groups were fed the B17-supplemented diet.

### 2.4. Experimental Design

The experimental design followed the methodology outlined by [18]. Briefly, ten pairs of *D. melanogaster* adults (3 days old) were collected from rearing vials and transferred into fresh 100 mL vials, one for each experimental group. Group 1 served as the control and was maintained on a standard control diet. Group 2 fed on the standard diet and was exposed to UVR. Group 3 was fed a diet supplemented with vitamin B17. Group 4 was fed a vitamin B17-supplemented diet and exposed to UVR. The UV-treated groups were exposed to radiation for one hour per day over a continuous seven-day period, following the protocol described by [12]. The assay was performed in ten replicates. All the following procedures were initiated after seven exposure periods.

### 2.5. Assessment of Vitamin B17 Effect

#### 2.5.1. Survival and Mortality Rate Analysis

To test the ameliorative effect of vitamin B17, if any, on survival, five pairs of adult *D. melanogaster* were collected randomly from each experimental group and transferred to fresh vials containing the same treatment medium. This was replicated three times. The number of dead flies in each group was recorded daily for two weeks.

#### 2.5.2. Fecundity Analysis

For the potential effect on fecundity, five females of *Drosophila* from each treatment were transferred to fresh, blue-colored food media with the same treatment the females came from and allowed to oviposit. After 24 h, the females were removed, and the number of eggs laid was counted. The experiment was replicated three times. Fecundity rate was calculated using the equation of [19].

#### 2.5.3. Developmental Time Analysis

The developmental time of their progeny (F1) generation in days (egg-adults) was monitored. The shape, color, length and width of the F1 pupal stage were also checked for phenotypic changes relative to the control using an Olympus BX61 microscope (Olympus Corporation, Tokyo, Japan). Additionally, the number of successfully emerged adults was recorded.

#### 2.5.4. Negative Geotaxis Assay

Male locomotor activity in each experimental group was evaluated using a modified negative geotaxis assay described by [20]. Briefly, 10 adult male flies from each experimental group were placed into a clean 100 mL glass cylinder. After a 10-min acclimation period, the flies were gently tapped to the bottom of the cylinder and allowed to climb. Their upward movement was recorded on video. Climbing speed was analyzed for each individual using ImageJ software (version 1.2) [21]. Flies that did not climb were assigned a speed of zero. The assay was conducted three times. Only male flies were used to eliminate variability associated with changes in female body weight during oogenesis and ovulation [22].

#### 2.5.5. Oxidative Stress Parameters Analysis

Oxidative stress parameters, including lipid peroxidation (LPO) measured as malondialdehyde level (MDA), as well as antioxidant enzymes, including superoxide dismutase (SOD) and catalase (CAT), were determined in the four experimental groups according to methods described previously [23,24]. In brief, flies (*n* = 50; 25 pairs) were homogenized in phosphate-buffered saline and then centrifuged at 11.180 g for 15 min at 4 °C to obtain the supernatant. MDA level was measured using the thiobarbituric acid reactive substances (TBARSs) assay. SOD activity was determined based on its ability to inhibit the autoxidation of pyrogallol, while CAT activity was measured by monitoring the decomposition rate of hydrogen peroxide (H_2_O_2_) spectrophotometrically. Measurements were repeated in triplicate.

#### 2.5.6. Apoptosis Assessment

To assess apoptosis, immunostaining was performed using an anti-cleaved Caspase-3 antibody. *Drosophila* heads (*n* = 10, 5 pairs) were fixed in 10% formalin. Following fixation, the samples were dehydrated through a graded ethanol series. Tissue sections were incubated overnight at 4 °C with a diluted (1:200) mouse anti-Caspase-3 monoclonal antibody (Clone 31A1067, Catalogue #MC0123, Medaysis, CA, USA), followed by incubation with a Horseradish Peroxidase (HRP) anti-polyvalent (ScyTek Laboratories, Logan, UT, USA) for 30 min at room temperature. Subsequently, the sections were treated with streptavidin for 10 min at room temperature and exposed to 3,3′-diaminobenzidine tetrahydrochloride (DAB) solution for 5–10 min, until the positive control displayed brown staining. Finally, the sections were counterstained with hematoxylin, rinsed under running water, dehydrated in ascending alcohol concentrations (70%, 80%, 90%, and 100%), and cleared in xylene for 5 min. Each staining batch included tonsil tissue as a positive control. The prepared sections were mounted and examined using a Leitz–Wetzlar photomicroscope (Ernst Leitz GmbH, Wetzlar, Germany) to evaluate immunostaining [25].

#### 2.5.7. Electron Microscopy Examination

The head regions of the flies (*n*= 20; 10 pairs) from each experimental group were fixed in a 2.5–3% glutaraldehyde solution prepared in 0.1 M cacodylate buffer (pH 7.2–7.4) at 4 °C for 2 h. Post-fixation was performed with 1% osmium tetroxide at 4 °C for an additional 1.5 h. The specimens were then dehydrated through a graded ethanol series (50%, 70%, 90%, 95%, and four changes of 100%), with each step lasting 15 min, followed by a 30-min dehydration step in acetone. Subsequently, the samples were embedded in epoxy resin using the Epoxy Embedding Medium Kit (Sigma, Darmstadt, Germany). For scanning electron microscopy (SEM), samples were dried using liquid CO_2_ and then gold-sputter coated using a SPI-Module™ Vac/Sputter 7 (Structure Probe, Inc., West Chester, PA, USA). Imaging was conducted with a JEOL JSM-5200 LV SEM (JEOL, Tokyo, Japan) at 20 kV. However, for transmission electron microscopy (TEM), ultra-thin sections were prepared using an RMC PT-XL PowerTome Ultramicrotome (Boeckeler Instruments, Inc., Tucson, AZ, USA), producing both semi-thin and ultra-thin slices. Ultra-thin sections, 70–90 nm thick, were stained with 2.5% uranyl acetate as the primary stain and counterstained with lead citrate, examined and imaged using a JEM-2100 transmission electron microscope (JEOL, Tokyo, Japan).

### 2.6. Statistical Analysis

Data were expressed as means ± standard deviation (SD). Because the experimental design consists of one categorical independent variable (treatment condition) with four levels (experimental groups), the comparison between the mean responses among these groups was conducted using one-way ANOVA, after checking the normality and homogeneity of data, followed by Tukey’s post hoc test; *p*-values < 0.05 were considered statistically significant. Microsoft Excel 365 (Microsoft Corporation, Redmond, WA, USA) and Minitab version 21 were used for data and statistical analysis.

## 3. Results

### 3.1. Ameliorative Effect of Vitamin B17 on Survival and Life History Traits

The results revealed that the mortality rates of *Drosophila* adults were null in all tested groups. However, the fecundity rate showed a highly significant difference among the experimental groups (Figure 2; One-way ANOVA; *p* = 0.001. Although flies exposed to UVR showed a slight increase in egg production compared to the control group, this difference was not statistically significant (*p* = 0.067), indicating that UVR alone had no substantial effect on fecundity. When vitamin B17-fed flies were exposed to UVR, the increase in egg laying became statistically significant (*p* = 0.001). Notably, the B17 group also showed higher fecundity than the control group, suggesting that the observed enhancement in reproductive output is primarily driven by vitamin B17 rather than UVR exposure.

A highly significant variation in developmental time (Figure 3) from egg to the adult stage among the four experimental groups (*p* = 0.001) was documented. The control group had the longest mean developmental time (9.33 days), while the B17-fed, UV-exposed group had the shortest (6.00 days). This pronounced reduction suggests a potential synergistic interaction between vitamin B17 and UVR in enhancing developmental progression. Groups exposed solely to B17-supplemented food or to UVR, with a control food regime, exhibited intermediate developmental times (8.00 and 7.00 days, respectively), indicating that each factor independently has a moderate effect on developmental rate compared to the control group.

Additionally, a highly significant difference in pupal dimensions (*p* = 0.001) was observed among the four experimental groups (Figure 4). UVR exposure led to a significant increase in both pupal length and width compared to the control group (*p* = 0.001), indicating a morphological response to radiation stress. However, when larvae were treated with vitamin B17 and concurrently exposed to UVR, pupal dimensions were comparable to those of the control group, with no significant difference in length (*p* = 0.623) and only a slight, yet statistically significant, difference in width (*p* = 0.04). These findings suggest that vitamin B17 may play a protective or modulatory role in mitigating the morphological impact of UVR during development.

As illustrated in Figure 5, there is a highly significant difference (*p* = 0.001) in the number of emerged *D. melanogaster* adults among the experimental groups. UVR exposure alone caused a significant reduction in adult emergence compared to the control (*p* = 0.04). In contrast, the group treated with vitamin B17 then exposed to UVR showed a significantly higher number of emerged adults than both the control and UV-only groups (*p* = 0.005). Notably, the group treated with vitamin B17 alone exhibited the highest number of adult emergences overall. These results suggest that the increased emergence in the B17+UV group is primarily due to the effect of vitamin B17 rather than UVR exposure.

### 3.2. Ameliorative Effect of Vitamin B17 on Locomotion

Statistical analysis showed a highly significant difference in climbing speed among experimental groups (*p* = 0.001). As illustrated in Figure 6, UVR exposure significantly impaired climbing speed compared to the control. In contrast, flies in the B17+UV group exhibited a higher climbing speed than the control group. Notably, the B17-only group recorded the highest overall climbing speed. These results indicate that vitamin B17 not only counteracts the adverse effects of UVR exposure on geotactic behavior but may also enhance climbing performance beyond control levels.

### 3.3. Biochemical Parameters

The data presented in Table 1 showed that exposure of *Drosophila* adults to UVR led to a marked increase in MDA levels compared to the control group. In contrast, flies that received vitamin B17 exhibited reduced MDA levels, even under UVR exposure. Moreover, UVR exposure resulted in a noticeable decline in the activities of the antioxidant enzymes SOD and CAT, whereas supplementation with vitamin B17 significantly restored their activities toward normal levels. These findings highlight the protective role of vitamin B17 against UVR-induced oxidative stress.

### 3.4. Ameliorative Effect of Vitamin B17 on Cellular Apoptosis

Immunohistochemical analysis showed a pronounced Caspase-3 signal throughout the optic lobe neuropils in flies exposed to UVR compared to controls (Figure 7A,B). Intense labeling was observed in the retina, lamina, medulla, lobula, and lobula plate, indicating that UVR exposure negatively impacted multiple visual processing centers. Although Caspase-3 signals were higher in UV-exposed flies fed with B17 than in non-UV-exposed B17-fed flies (Figure 7C,D), both the intensity and distribution were lower than in flies on the control diet without vitamin supplementation. This suggests that the vitamin may help protect against UVR-induced damage.

### 3.5. Ultrastructure Examination

#### 3.5.1. Scanning Electron Microscopy

Scanning electron microscopy of the compound eyes revealed that the control group exhibited typical, healthy morphology, characterized by uniform hexagonal ommatidia with well-defined boundaries and regularly spaced interommatidial bristles (Figure 8A). In contrast, the UV-exposed group displayed marked disorganization of ommatidia and disruption of surface integrity (Figure 8B). Flies fed with B17 maintained largely preserved eye architecture, with intact ommatidia and orderly bristle arrangement (Figure 8C). Notably, the B17-fed UV-exposed group showed structurally organized ommatidia with aligned bristles (Figure 8D), suggesting that B17 supplementation partially mitigates UVR-induced morphological damage.

#### 3.5.2. Transmission Electron Microscopy

Transmission electron microscope analysis revealed distinct differences in ommatidial ultrastructure and integrity among the various treatment groups. In the control group (Figure 9A), showing normal rhabdom architecture, with seven rhabdomeres surrounded by pigment cells, a stalked rhabdomere with consistent direction. The inset highlights a normal rhabdomere with well-organized microvilli and sub-microvilli cisternae. In the UV-treated group (Figure 9B), ommatidial structure exhibited severe abnormalities, including disrupted microvilli and irregularly shaped rhabdoms with evidence of vacuolation, indicative of cellular damage associated with UVR exposure. In the B17-fed group (Figure 9C), rhabdomere morphology remained largely intact. Most rhabdoms preserved their organized microvilli, similar to those in the control group. The overall cellular structure appeared normal, suggesting that vitamin B17 maintains cellular homeostasis. In the B17-UV-exposed group (Figure 9D), partial recovery of rhabdom structure was observed. While some rhabdomeres still showed signs of swelling and disrupted microvilli, many appeared more structured and less vacuolated than in the UV-only group.

In the control group (Figure 10A), the mitochondria exhibited intact membranes and clearly defined cristae, while the cytoplasmic matrix appeared normal and uniform, indicating overall cellular health. In contrast, the UV-exposed flies (Figure 10B) showed marked signs of degeneration. Mitochondria appeared swollen and vacuolated, and the cytoplasm showed vacuolization, reflecting UVR-induced cellular damage. The vitamin B17-fed flies (Figure 10C) maintained a highly organized cellular architecture. Numerous elongated mitochondria with intact cristae were observed, and endoplasmic reticulum structures were clearly visible. The nuclei were well-defined, with an intact nuclear membrane and visible chromatin, indicating preserved cellular integrity and potential enhancement of mitochondrial structure. In the B17+UV-exposed group (Figure 10D), partial restoration of ultrastructure was observed compared to the UV-only group. Although some mitochondria appeared mildly swollen, the extent of cytoplasmic vacuolization was less, indicating that vitamin B17 conferred a degree of protection against UVR-induced cellular damage.

## 4. Discussion

The present study provides insight to clarify whether vitamin B17 exerts a genuine protective role against UVR-induced damage in *D. melanogaster*, thereby contributing to the ongoing debate regarding its proposed benefits.

The findings revealed that adult flies exhibited no mortality across all four treatment groups. This finding is consistent with previous studies [26], which reported that UVA radiation did not induce mortality in *D. melanogaster*. Furthermore, statistical analysis revealed that UVR exposure alone had no significant effect on fecundity. However, a significant increase in egg production was observed in adult females supplemented with vitamin B17 and subsequently exposed to UVR compared to all other groups. This result suggests that the observed enhancement in reproductive output is primarily attributed to vitamin B17. It is plausible that vitamin B17 enhances oogenesis or stimulates ovarian function, particularly under conditions of environmental stress such as UVR exposure. This interpretation is supported by [16], who demonstrated that guava extracts exerted a protective effect on the reproductive capacity of *D. melanogaster* under UV stress.

Previous studies have demonstrated that UVR exposure can interfere with normal insect development. For example, ref. [27] reported that UVR exposure disrupted larval development and delayed adult eclosion. While ref. [28] found that UVR exposure significantly inhibited larval development and adult emergence in *Dialeurodes citri*. The present study also revealed a significant reduction in developmental time in the UV, B17, and B17+UV groups compared to the control group. Notably, the shortest developmental duration was observed in the B17+UV group. The reduction in developmental time following UVR exposure may indicate a stress-induced acceleration of metamorphosis, potentially as an adaptive strategy to environmental stress, enabling faster transition to adulthood. Meanwhile, the further shortening of developmental time in the B17+UV group suggests that B17 may enhance this stress-response mechanism or promote more efficient development despite UVR-induced challenges. This hypothesis was consistent with [29], who indicated that stresses significantly influence the development of insects.

Interestingly, both UVR exposure and B17 supplementation independently increased pupal size compared to the control. While this stimulatory effect may reflect a hormetic response to mild stress, the outcome was markedly different when B17 was combined with UV treatment. In this case, pupal size returned to control levels, indicating that B17 shifted from a stimulatory role under normal conditions to a protective and modulatory role under stress. Such preconditioning may involve the activation of antioxidant defenses or DNA repair pathways that buffer the developmental disruptions induced by UVR exposure. These results are consistent with [16], who reported a modulatory effect of guava on reducing UV-related morphological changes in *Drosophila.*

UVR exposure decreased adult emergence, consistent with the findings of [18] regarding UVC effects on *D. melanogaster*. In contrast, the combined treatment with UV and B17 significantly enhanced adult emergence, possibly because B17 supports physiological processes critical for successful development.

The present findings also demonstrated that UVR exposure significantly reduced climbing speed, likely due to neuromuscular impairment or oxidative stress affecting neural circuits involved in geotaxis. These results align with those of [14], who reported that UVR slows the movement of *D. melanogaster*. However, the treatment with B17 (B17+UV group) reversed this effect, enhancing geotactic activity beyond the control level. The highest speed was observed in the B17-only group, suggesting possible neuroprotective or metabolic benefits of B17.

At the biochemical level, oxidative stress induced by UVR exposure is typically characterized by elevated ROS levels. In insects, the antioxidant defense system, including key enzymes such as SOD, CAT, and peroxidase, plays a critical role in mitigating oxidative damage. Variations in the activity of these enzymes have been reported in multiple insect species following UVR exposure [10,30]. Here, exposure of *D. melanogaster* to UVR resulted in a significant increase in lipid peroxidation and a marked reduction in the activities of antioxidant enzymes, SOD and CAT, indicating the onset of oxidative stress and cellular damage, consistent with results of [10,18]. Supplementation with vitamin B17 led to a notable restoration of redox balance by lowering LPO and reinstating SOD and CAT activity, suggesting that vitamin B17 may act through direct ROS scavenging or by upregulating antioxidant enzyme expression. These explanations are consistent with findings by [31], who emphasized the role of natural compounds and dietary supplements in mitigating oxidative stress.

The findings also demonstrated that exposure to UVR caused significant morphological and ultrastructural changes in the compound eye of *D. melanogaster*. These alterations included corneal deformation and a marked reduction in ocular sensilla, consistent with [32], who observed UVR-induced photoreceptor damage across multiple insect species. Notably, these harmful effects were significantly mitigated by vitamin B17 supplementation, likely due to its role in stabilizing cellular membranes and modulating inflammation [32]. In support of this interpretation, the authors demonstrated that α-tocopherol and its derivative, α-tocopherol acetate, can modulate UVB-induced apoptosis and inhibit NF-κB activation, a key transcription factor involved in oxidative stress and inflammatory responses. Similarly, Ambagaspitiy et al. [9] reported that vitamin D exerts photoprotective effects by facilitating the repair of cyclobutane pyrimidine dimers (CPDs), mitigating oxidative stress, and reducing inflammation. Our findings further demonstrate that the protective effects of B17 extend beyond the preservation of tissue structures to the cellular energy machinery, as supplementation effectively maintained mitochondrial integrity under UV exposure. This suggests that B17 not only protects visible tissue architecture but also safeguards critical subcellular organelles, highlighting its broad cytoprotective capacity and its role in maintaining cellular homeostasis under stress. Building on these compelling results, future studies using specific mitochondrial markers and functional assays will provide deeper insight into the mechanisms of B17’s protective actions and enhance our understanding of its role in supporting mitochondrial function and overall cellular resilience.

From a translational perspective, the current findings underscore the potential of vitamin B17 (amygdalin) to alleviate UVR-induced biological and physiological damage in *D. melanogaster*. These results expand the potential applications of B17, supporting its reconsideration not only in light of its commercially promoted role as an anticancer agent but also as a prospective neuroprotective compound relevant to ocular health, especially under prolonged UVR exposure. However, ongoing concerns about B17’s metabolism, particularly the potential release of toxic byproducts such as hydrogen cyanide, underscore the need for comprehensive pharmacokinetic and toxicological assessments before any therapeutic use is pursued. Future research should focus on clarifying the molecular mechanisms underlying its protective action, exploring potential synergistic effects in combination therapies, and assessing its protective capacity across different tissues using other neuro-ophthalmological models.

## Figures and Tables

**Figure 1 insects-16-01238-f001:**
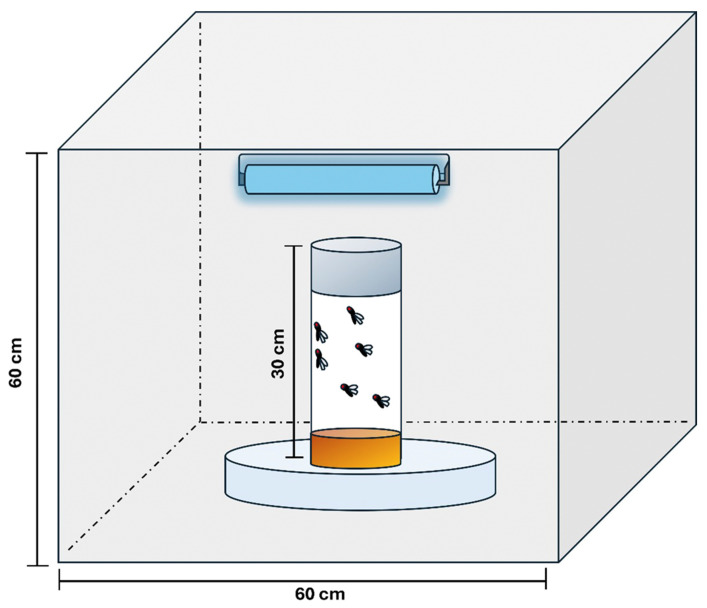
Schematic diagram of the UVR exposure setup. Adult *Drosophila melanogaster* were placed inside closed bottles positioned on an inverted dish at a height of 30 cm from a 100 W UV fluorescent lamp. The lamp was mounted at the top of a reflective chamber (60 × 60 × 60 cm^3^) to ensure uniform UVR exposure.

**Figure 2 insects-16-01238-f002:**
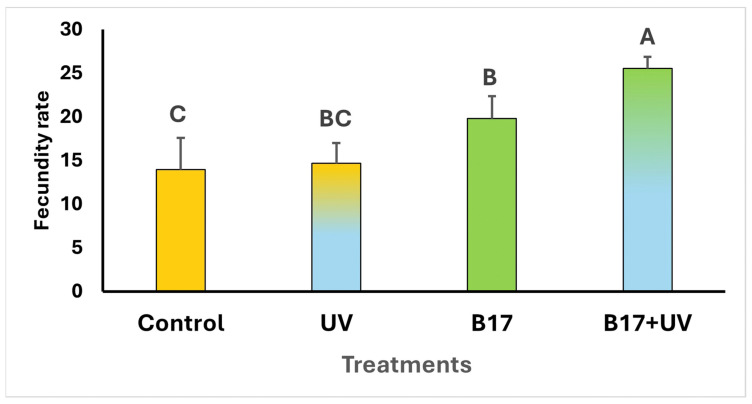
Fecundity rate (the mean number of eggs laid per 5 females *Drosophila melanogaster* ± SD) under four treatment conditions: control non-UV-exposed flies, control UV-exposed flies, non-UV-exposed flies fed on vitamin B17-supplemented food media, and UV-exposed flies fed on vitamin B17-supplemented food media. Different letters above bars indicate statistically significant differences between groups (Tukey’s test, *p* < 0.05). The letters above the bars indicate the statistical ranking of the measured means. Specifically, the letter ‘A’ corresponds to the highest value, followed by ‘B’ and ‘C’, with the letters ordered from the highest to the lowest mean (A > B > C). Error bars represent standard deviation.

**Figure 3 insects-16-01238-f003:**
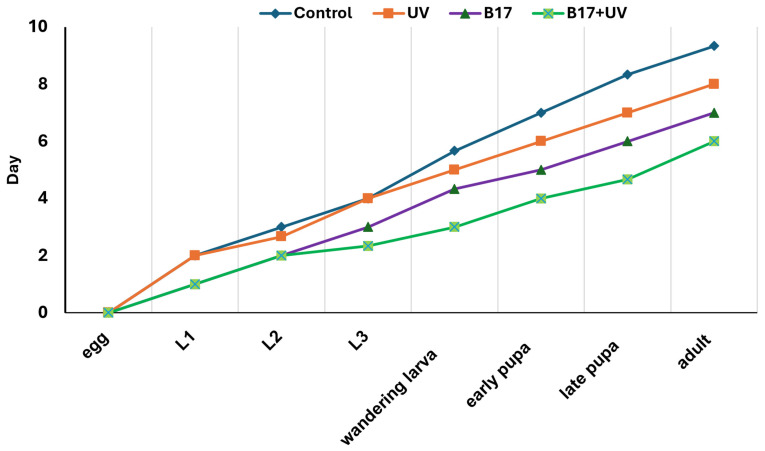
Developmental period (days) from egg to adult stage *Drosophila melanogaster* under four treatment conditions: control non-UV-exposed flies, control UV-exposed flies, non-UV-exposed flies fed on Vitamin B17-supplemented food media, and UV-exposed flies fed on vitamin B17-supplemented food media. Statistically significant differences between groups at *p* < 0.05 (Tukey’s test, *p* < 0.05).

**Figure 4 insects-16-01238-f004:**
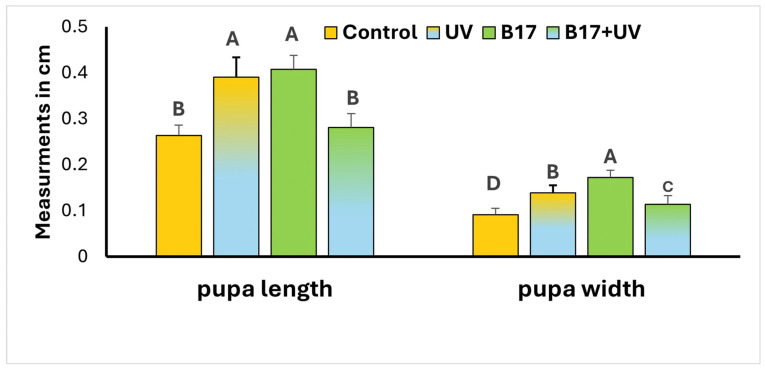
Measurement of *Drosophila melanogaster* pupal stage length and width (Mean ± SD) under four treatment conditions: control non-UV-exposed flies, control UV-exposed flies, non-UV-exposed flies fed on vitamin B17-supplemented food media, and UV-exposed flies fed on vitamin B17-supplemented food media. Different letters above bars within the same parameter indicate statistically significant differences between groups (Tukey’s test, *p* < 0.05). The letters above the bars within the same parameter indicate the statistical ranking of the measured means. Specifically, the letter ‘A’ corresponds to the highest value, followed by ‘B’, ‘C’, and ‘D’, with the letters ordered from the highest to the lowest mean (A > B > C > D). Error bars represent standard deviation.

**Figure 5 insects-16-01238-f005:**
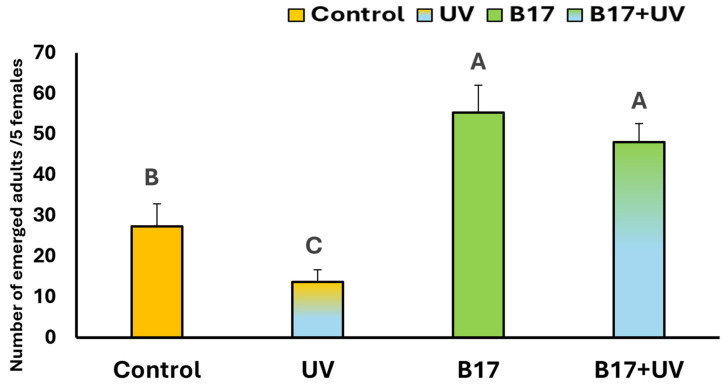
Number of emerged *Drosophila* adults per 5 females (Mean ± SD) under four treatment conditions: control non-UV-exposed flies, control UV-exposed flies, non-UV-exposed flies fed on vitamin B17-supplemented food media, and UV-exposed flies fed on vitamin B17-supplemented food media. Different letters above bars indicate statistically significant differences between groups (Tukey’s test, *p* < 0.05). The letters above the bars indicate the statistical ranking of the measured means. Specifically, the letter ‘A’ corresponds to the highest value, followed by ‘B’, and ‘C’, with the letters ordered from the highest to the lowest mean (A > B > C). Error bars represent standard deviation.

**Figure 6 insects-16-01238-f006:**
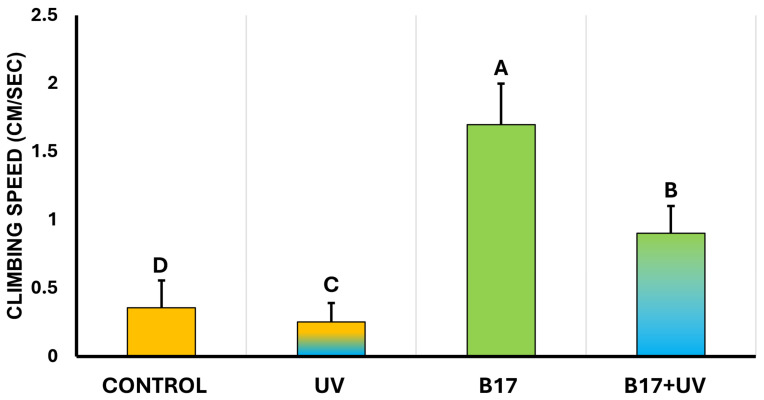
Climbing speed of *Drosophila melanogaster* adults (cm /sec; *n* = 10) (Mean ± SD) under four treatment conditions: control non-UV-exposed flies, control UV-exposed flies, non-UV-exposed flies fed on vitamin B17-supplemented food media, and UV-exposed flies fed on Vitamin B17-supplemented food media. Different letters above bars indicate statistically significant differences between groups (Tukey’s test, *p* < 0.05). The letters above the bars indicate the statistical ranking of the measured means. Specifically, the letter ‘A’ corresponds to the highest value, followed by ‘B’, ‘C’, and ‘D’, with the letters ordered from the highest to the lowest mean (A > B > C > D). Error bars represent standard deviation.

**Figure 7 insects-16-01238-f007:**
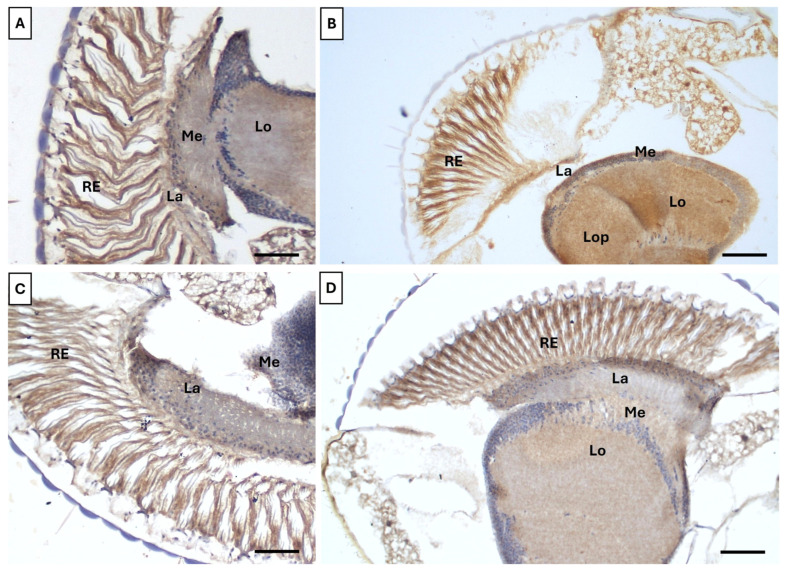
Adult *Drosophila melanogaster* compound eye immunohistochemically labeled with anti-caspase3 antibody showing the ameliorative effect of vitamin B17 on the apoptotic effect of exposure to ultraviolet radiation for 1h daily for a week. (**A**), control non-UV-exposed flies. (**B**), control UV-exposed flies. (**C**), non-UV-exposed flies fed on vitamin B17-supplemented food media. (**D**), UV-exposed flies fed on vitamin B17-supplemented food media. Lo, lobula. RE, retina. La, lamina Me, medulla. Scale bar = 20 µm.

**Figure 8 insects-16-01238-f008:**
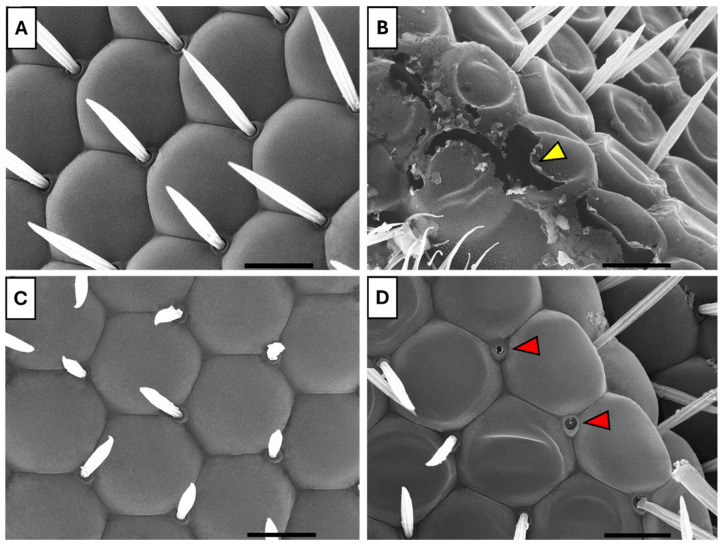
Scanning electron microscope images showing the ameliorative effect of vitamin B17 on the adverse impact of exposure to ultraviolet radiation for 1h daily for a week that appeared on the compound eyes facets of adult *Drosophila melanogaster*. (**A**), control non-UV-exposed flies. (**B**), control UV-exposed flies. (**C**), non-UV-exposed flies fed on vitamin B17-supplemented food media. (**D**), UV-exposed flies fed on vitamin B17-supplemented food media. Yellow arrowhead refers to damaged corneal tissue resulting from exposure to UV. The red arrowhead indicates the loss of some sensilla. Scale bar = 10 µm.

**Figure 9 insects-16-01238-f009:**
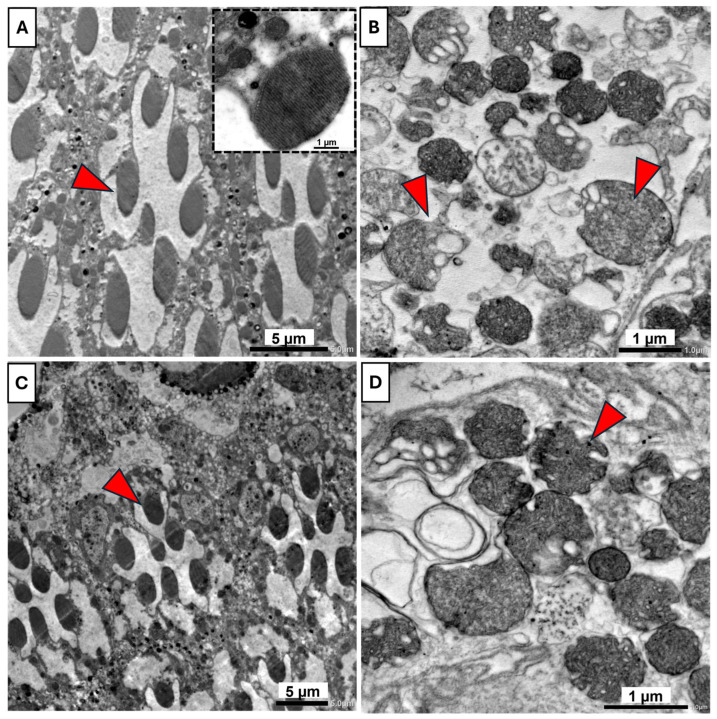
Transmission electron microscopy photomicrographs of the compound eye of *Drosophila melanogaster* adults showing the ameliorative effect of vitamin B17 on the adverse impact of exposure to ultraviolet radiation for 1h daily for a week. (**A**), the typical architecture of rhabdom contains 7 rhabdomeres (red arrowheads) arranged in a normal manner, and the inter-rhabdomeres space appeared between rhabdomeres of control non-UV exposed flies. The inset shows parallel-arranged microvilli of a rhabdomere. (**B**), the rhabdom tissue was damaged, lost its regular shape and the normal arrangement of the 7 rhabdomeres; additionally, typical-arranged microvilli were lost, and vacuolization appeared in control UV-exposed flies. (**C**), typical rhabdom architecture appeared in non-UV-exposed flies fed on vitamin B17-supplemented food media. (**D**), Damaged rhabdom tissue appeared in UV-exposed flies fed on vitamin B17-supplemented food media. Scale bars are as indicated in each image.

**Figure 10 insects-16-01238-f010:**
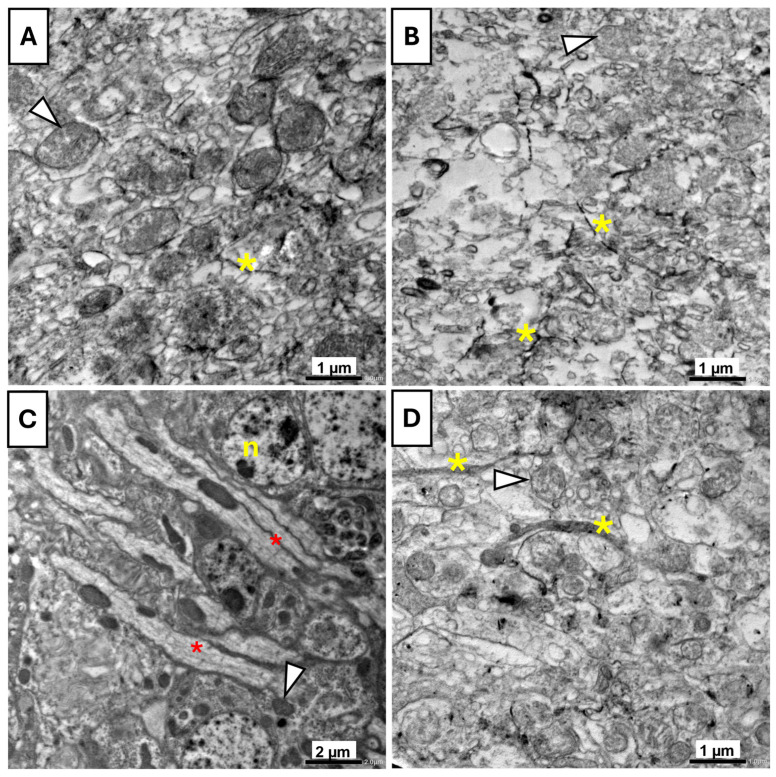
Transmission electron microscopy photomicrographs of the brain tissue of *Drosophila melanogaster* adults showing the ameliorative effect of vitamin B17 on the adverse impact of exposure to ultraviolet radiation for 1h daily for a week. (**A**), control non-UV-exposed flies. (**B**), control UV-exposed flies. (**C**), non-UV-exposed flies fed on vitamin B17-supplemented food media. (**D**), UV-exposed flies fed on vitamin B17-supplemented food media. Yellow asterisks indicate axons, red asterisks indicate myelin-like structures, and arrowheads refer to mitochondria. n, nucleus. Scale bars are as indicated in each image.

**Table 1 insects-16-01238-t001:** MDA level and antioxidant enzyme activities (SOD and CAT) in *Drosophila melanogaster* post exposure to different UV radiation and vitamin B17 treatment conditions.

Treatment	MDA (nmol/mg Protein)	SOD (U/mg Protein)	CAT (U/mg Protein)
Control	3.81 ± 0.45 ^c^	5.83 ± 0.55 ^a^	96.61 ± 4.15 ^a^
UV	9.46 ± 0.91 ^a^	3.06 ± 0.61 ^b^	54.37 ± 3.25 ^c^
B17	3.16 ± 0.30 ^d^	6.12 ± 0.30 ^a^	108.55 ± 3.67 ^a^
B17+UV	5.40 ± 0.56 ^b^	5.29 ± 0.56 ^a^	78.61 ± 3.86 ^b^
*p* value	0.001	0.005	0.008

MDA: Malondialdehyde; SOD: superoxide dismutase; CAT: catalase; U: international unit (µmol/minute/mL) (Mean ± SD). The means with the same superscript letters within the same column are not significantly different at *p* > 0.05. The superscript letters indicate the statistical ranking of the measured means within the same column. Specifically, the letter ‘a’ corresponds to the highest value, followed by ‘b’,‘ c’, and ‘d’ with the letters ordered from the highest to the lowest mean (a > b > c > d).

## Data Availability

The original contributions presented in this study are included in the article. Further inquiries can be directed to the corresponding author.

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
