# Peer review of "Evaluation of the Protective Effect of Vitamin B17 Against the Potential UV Damage Using Drosophila as a Model"

_insects, 2025, doi:10.3390/insects16121238_

Round 1

Reviewer 1 Report

Comments and Suggestions for Authors

Drosophila has been used as the model for studying VB17 to protect against the harmful effect of UVR. In this study, authors found VB17 exerted a protective effect against UV-induced adverse consequences in adult flies, highlighting the benefits of VB17. However, some major revisions are required before this paper can be accepted.

Major comments:

  1. Authors showed flies fed VB17-supplemented food prior to UVR exposure displayed markedly higher adult emergence rates, improved climbing ability and shortened developmental time compared to UV exposed flies on standard food. At cellular level, VB17 supplementation reduced Caspase-3 activation, preserved the structural integrity of compound eyes and mitochondria. How to explain the mechanism between these effects.
  2. The mechanism of VB17 to UVR is still unclear. Authors should at least analysis the gene expression related to VB17 or other related process between VB17-supplement and the control group.
  3. Authors showed VB17 supplementation preserved the structural integrity of mitochondria by using TEM assay. However, the structure of mitochondrion should be further determined by using ATP5A or Tom20 antibodies. Moreover, the function of mitochondrion is needed to be determined.
  4. There are still some grammatical errors in the article, the author needs to check carefully and revise it.

Author Response

Drosophila has been used as the model for studying VB17 to protect against the harmful effect of UVR. In this study, authors found VB17 exerted a protective effect against UV-induced adverse consequences in adult flies, highlighting the benefits of VB17. However, some major revisions are required before this paper can be accepted.

Major comments:

  1. Authors showed flies fed VB17-supplemented food prior to UVR exposure displayed markedly higher adult emergence rates, improved climbing ability and shortened developmental time compared to UV exposed flies on standard food. At cellular level, VB17 supplementation reduced Caspase-3 activation, preserved the structural integrity of compound eyes and mitochondria. How to explain the mechanism between these effects.

Response: We thank the reviewer for this valuable comment. The protective effects of VB17 observed in our study can be mechanistically interpreted based on the parameters we directly measured. UV radiation is known to induce oxidative stress and apoptosis, yet flies receiving VB17 supplementation showed a marked reduction in Caspase-3 activation, indicating suppression of UV-triggered apoptotic pathways. This reduced apoptotic response was accompanied by clear preservation of mitochondrial structure, as demonstrated by our ultrastructural analyses. Since mitochondrial damage is a major consequence of reactive oxygen species accumulation after UV exposure, the maintenance of normal mitochondrial morphology in VB17-treated flies suggests that VB17 either enhances cellular antioxidant capacity or stabilizes mitochondrial membranes, thereby reducing oxidative injury. These cellular protective effects were reflected at the tissue and organismal levels. VB17-fed flies exhibited preserved compound eye architecture, consistent with reduced cell death and improved structural integrity under UV stress.

  1. The mechanism of VB17 to UVR is still unclear. Authors should at least analysis the gene expression related to VB17 or other related process between VB17-supplement and the control group.

Response: We appreciate the reviewer’s valuable suggestion regarding the analysis of gene expression to further elucidate the mechanism of VB17 under UVR stress. We fully agree that molecular profiling would provide deeper mechanistic insights; however, our current study was designed as a first-level investigation focusing primarily on physiological and cellular outcomes. Despite the absence of gene-expression analysis, our results already highlight several mechanistic clues. Nevertheless, we fully acknowledge the importance of this direction. In future studies, we plan to incorporate gene expression profiling and other molecular analyses to further elucidate the pathways involved and strengthen our mechanistic understanding of VB17’s effects

  1. Authors showed VB17 supplementation preserved the structural integrity of mitochondria by using TEM assay. However, the structure of mitochondrion should be further determined by using ATP5A or Tom20 antibodies. Moreover, the function of mitochondrion is needed to be determined.

Response: We thank the reviewer for this insightful comment regarding mitochondrial characterization. Although immunolabeling with mitochondrial markers such as ATP5A or Tom20 would indeed offer complementary information on specific mitochondrial compartments, this approach was beyond the scope of the current study design, which focused on establishing foundational phenotypic and ultrastructural evidence of VB17-mediated protection. In the present study, transmission electron microscopy (TEM) was our primary tool for assessing mitochondrial integrity because it provides direct visualization of ultrastructural features such as cristae organization, membrane continuity, and overall morphology—hallmarks that are well-established indicators of mitochondrial damage under UV stress. Our TEM data revealed clear preservation of mitochondrial structure in VB17-supplemented flies compared to the UV-exposed control group. Nevertheless, in response to the reviewer’s valuable suggestion, we have added a paragraph in the Discussion section explicitly acknowledging this limitation and outlining our plan to incorporate both mitochondrial protein markers (ATP5A/Tom20) and functional assays in future work (lines 450-458).

There are still some grammatical errors in the article, the author needs to check carefully and revise it.

Response: We appreciate the reviewer’s comment regarding language and grammatical accuracy. We have carefully revised the manuscript and performed a thorough English language edit to ensure clarity, correct grammar, and readability throughout the paper.

Reviewer 2 Report

Comments and Suggestions for Authors

The authors present a study investigating the potential protective role of a controversial compound, vitamin B17 (amygdalin), against UV-induced damage. The utilization of the Drosophila melanogaster model is appropriate for such a screening study. The manuscript would, however, benefit from substantial revisions to strengthen the methodological description, statistical rigor, and the depth of the biological discussion. The following comments are offered to assist the authors in improving their work.

Introduction

The hypothesis driving the study should be more explicitly stated. Furthermore, the introduction would be strengthened by a clearer elaboration on the known or proposed biochemical role of vitamin B17. A mechanistic premise for how it could confer resistance to Ultraviolet Radiation is currently lacking. The authors should briefly discuss the compound's nature as a cyanogenic glycoside and any theoretical pathways through which it, or its metabolites, might interact with cellular stress responses, such as modulating oxidative stress.

Materials and Methods

The methodology requires significant expansion to ensure reproducibility. Key details are missing.

Please specify the sex of the flies used in each experiment.

A subsection dedicated to Vitamin B17 Treatment Protocol is necessary. It must include:

  • The commercial source and purity of the vitamin B17 (amygdalin/laetril) used.
  • The developmental stage (e.g., larvae, adults) and age at which treatment commenced.
  • The concentration of B17 tested and the solvent used for preparation (e.g., DMSO, ethanol, water).
  • The precise method of incorporation into the fly medium.
  • The temporal relationship between B17 treatment and UVR exposure (e.g., pre-treatment, co-treatment, post-treatment).

The exact duration and dosage (e.g., J/m²) of UV exposure must be specified.

A separate subsection describing the protocols for tracking and analyzing survival and mortality rates is required.

The subsection "2.2. Effect of Ultraviolet Radiation Exposure" is currently too broad. It is recommended to restructure it into logically distinct subsections for improved clarity, for example:

  • Ultraviolet Radiation Exposure Setup
  • Experimental Groups and Design
  • Assessment of Vitamin B17 Effect
  • Survival and Mortality Rate Analysis
  • Fecundity Analysis
  • Developmental Time Analysis
  • Negative Geotaxis Assay
  • Oxidative Stress Parameters Analysis
  • Apoptosis Assessment

Results

The results of the survival and mortality rate analysis, requested in the M&M section, must be presented clearly in this section.

The use of a one-way ANOVA is not optimal for the experimental design. Since the study involves two primary factors (Vitamin B17 and UV exposure) a two-way ANOVA is the statistically appropriate method. This would allow you to assess not only the main effects of each factor but also the critical interaction between B17 and UV, which is central to your hypothesis of a "protective effect." A re-analysis of the data using a two-way ANOVA is strongly recommended.

All figures require comprehensive legends. Each element must be clearly labeled to identify the experimental groups. Any letter designations (e.g., A, B, C, D) indicating statistical groupings must be explicitly defined. For each comparison shown, the reader must be able to discern which specific groups are being compared and the associated significance level.

Discussion

The suggestion in lines 367-370 regarding "stress-induced acceleration of metamorphosis" as a potential adaptive strategy requires stronger support. This concept should be argued with references to established literature on developmental plasticity and stress responses. For instance, you could cite studies discussing how environmental stressors can influence developmental timing and life-history traits in Drosophila and other models . This will ground your interpretation in a broader evolutionary context.

The discussion should extensively address the fundamental nature of vitamin B17 and how the observed effects might align with its known cellular actions. It is crucial to note that "vitamin B17" is a misnomer. It is not a recognized vitamin but a cyanogenic glycoside called amygdalin. The discussion must engage with the controversial status of this compound. You should discuss its metabolism, which can release cyanide, and consider whether the observed effects could be linked to low-level chemical hormesis rather than a vitamin-like action.

Author Response

The authors present a study investigating the potential protective role of a controversial compound, vitamin B17 (amygdalin), against UV-induced damage. The utilization of the Drosophila melanogaster model is appropriate for such a screening study. The manuscript would, however, benefit from substantial revisions to strengthen the methodological description, statistical rigor, and the depth of the biological discussion. The following comments are offered to assist the authors in improving their work.

 Introduction

The hypothesis driving the study should be more explicitly stated. Furthermore, the introduction would be strengthened by a clearer elaboration on the known or proposed biochemical role of vitamin B17. A mechanistic premise for how it could confer resistance to Ultraviolet Radiation is currently lacking. The authors should briefly discuss the compound's nature as a cyanogenic glycoside and any theoretical pathways through which it, or its metabolites, might interact with cellular stress responses, such as modulating oxidative stress.

Response:

We appreciate the reviewer’s comment. In response, the Introduction has been strengthened to explicitly state the hypothesis driving the study and to clarify the known and proposed biochemical roles of vitamin B17. Specifically, we have added paragraphs describing B17 as a cyanogenic glycoside, detailing its chemical composition and the metabolites it produces, and discussing the theoretical mechanisms through which it may modulate oxidative stress, preserve mitochondrial and tissue integrity, and influence apoptosis. These additions were in lines 44-59

Materials and Methods

The methodology requires significant expansion to ensure reproducibility. Key details are missing.

Please specify the sex of the flies used in each experiment.

Response: We appreciate the reviewer’s comment. In response, the Materials and Methods section has been significantly expanded to provide all necessary details to ensure reproducibility. All additions have been highlighted in red color.

A subsection dedicated to Vitamin B17 Treatment Protocol is necessary. It must include:

  • The commercial source and purity of the vitamin B17 (amygdalin/laetril) used.
  • The developmental stage (e.g., larvae, adults) and age at which treatment commenced.
  • The concentration of B17 tested and the solvent used for preparation (e.g., DMSO, ethanol, water).
  • The precise method of incorporation into the fly medium.
  • The temporal relationship between B17 treatment and UVR exposure (e.g., pre-treatment, co-treatment, post-treatment).

 Response: We appreciate the reviewer’s comment. All the requested details have been added to the manuscript (lines 111-118).

The exact duration and dosage (e.g., J/m²) of UV exposure must be specified.

Response: We appreciate the reviewer’s comment. The exact duration and lamp specifications for UV exposure, including the estimated UV dose, have been specified in the Materials and Methods section lines 108-109.

A separate subsection describing the protocols for tracking and analyzing survival and mortality rates is required.

Response: Done.

The subsection "2.2. Effect of Ultraviolet Radiation Exposure" is currently too broad. It is recommended to restructure it into logically distinct subsections for improved clarity, for example:

  • Ultraviolet Radiation Exposure Setup
  • Experimental Groups and Design
  • Assessment of Vitamin B17 Effect
  • Survival and Mortality Rate Analysis
  • Fecundity Analysis
  • Developmental Time Analysis
  • Negative Geotaxis Assay
  • Oxidative Stress Parameters Analysis
  • Apoptosis Assessment

Response: We appreciate the reviewer’s comment. This part was restructured into the suggested subsections.

Results

The results of the survival and mortality rate analysis, requested in the M&M section, must be presented clearly in this section.

Response: We thank you for your valuable comment. We would like to clarify that the survival and mortality results are indeed presented in the Results section. Specifically, we stated that all groups showed 0% mortality (100% survival) during the entire experimental period line 204.

The use of a one-way ANOVA is not optimal for the experimental design. Since the study involves two primary factors (Vitamin B17 and UV exposure) a two-way ANOVA is the statistically appropriate method. This would allow you to assess not only the main effects of each factor but also the critical interaction between B17 and UV, which is central to your hypothesis of a "protective effect." A re-analysis of the data using a two-way ANOVA is strongly recommended.

Response: We thank the reviewer for this valuable suggestion regarding the use of two-way ANOVA. We would like to clarify that the experimental design included four groups, with each parameter measured individually for each group. So, there is no multiply factors that required two-way Anova. Also, we like to mention that the fourth group (B17+UV) specifically represents the interaction between UV exposure and the vitamin treatment. Nonetheless, if the reviewer prefers that we apply a two-way ANOVA, we have no objection, but we kindly request further guidance on the appropriate model specification to ensure accurate analysis.

All figures require comprehensive legends. Each element must be clearly labeled to identify the experimental groups. Any letter designations (e.g., A, B, C, D) indicating statistical groupings must be explicitly defined. For each comparison shown, the reader must be able to discern which specific groups are being compared and the associated significance level.

Response: We have carefully reviewed all figure legends, and all the requested information is now clearly included. Each element in the figures is labeled to indicate the corresponding experimental group. Letter designations (A, B, C, D) used for statistical groupings are explicitly defined. All comparisons between groups, along with the associated significance levels, are clearly indicated. No data were missing; any details that were previously unclear have been added to the legends for full clarity. 

Discussion

The suggestion in lines 367-370 regarding "stress-induced acceleration of metamorphosis" as a potential adaptive strategy requires stronger support. This concept should be argued with references to established literature on developmental plasticity and stress responses. For instance, you could cite studies discussing how environmental stressors can influence developmental timing and life-history traits in Drosophila and other models . This will ground your interpretation in a broader evolutionary context.

Response: Thank you for this insightful comment. We agree that interpreting the observed acceleration of metamorphosis under environmental stress as an adaptive strategy requires stronger support. To address this, we have expanded our discussion to include the framework of developmental plasticity and stress responses. Lines 404-405.

The discussion should extensively address the fundamental nature of vitamin B17 and how the observed effects might align with its known cellular actions. It is crucial to note that "vitamin B17" is a misnomer. It is not a recognized vitamin but a cyanogenic glycoside called amygdalin. The discussion must engage with the controversial status of this compound. You should discuss its metabolism, which can release cyanide, and consider whether the observed effects could be linked to low-level chemical hormesis rather than a vitamin-like action.

Response: We appreciate the reviewer’s comment. A comprehensive section detailing the nature of amygdalin, its metabolism, and known mechanisms of action has already been added to the Introduction  lines 44-59 . To avoid redundancy, we have referred to this section in the Discussion, providing only a brief summary of its relevance to our findings.

Reviewer 3 Report

Comments and Suggestions for Authors

Authors studied protective effects of vitamin B17 against potential UV damage using Drosophila as model subject. That is not novel, but of interest also for human UV protection. The results combine life-cycle data with biochemical, histological and cytological observations. The results confirm vitamin B17 as modulator against UV stress, even when it is known to be toxic to other organisms.

Presentation of the results as well as text editing need some revision (see below).

An interesting study on the use of vitamin B17 against UV damage by using Drosophila as a model. 

-citations in the text as well as in the References must be given according to MDPI Authors' Instructions

  • line 11 and others: all genus and species names must be written in italics, also in the References
  • line 65: authors should give reasons, why they used Drosophila as model organism
  • line 113: there is no section 2.5 in the manuscript
  • line 138: centrifugation values must be presented in g, not in rpm
  • line 143: why Caspase here in uppercase letters?
  • line 146: v/v?
  • line 147: explain acronym for horse reddish peroxidase here
  • line 168: authors also used scanning EM, but did not mention this in Materials and Methods
  • legend to Fig.2: say that eggs were laid by 5-day old females within 24 hours
  • legend to figure 3: no statistics is shown in the figure 
  • line 216: pupal
  • Fig. 6: data must be presented in columns. You cannot connect the results with line
  • line 250: data are shown in Table 2, not in Table 1. Table 1 can be deleted; the explanatiosn were given before.
  • Table 2: better say, which are significantly different
  • Fig. 7: it would have been better using graphs with same image sections for better comparison
  • line 277: why uppercase letters here? Insert a space between number and unit (see also Fig. 8)
  • Fig. 10 C: use n or N, but not mixed
  • line 402: better say MDA level
  • References must be adapted to MDPI Authors' Instructions (see above).

Author Response

Authors studied protective effects of vitamin B17 against potential UV damage using Drosophila as model subject. That is not novel, but of interest also for human UV protection. The results combine life-cycle data with biochemical, histological and cytological observations. The results confirm vitamin B17 as modulator against UV stress, even when it is known to be toxic to other organisms.

Presentation of the results as well as text editing need some revision (see below).

An interesting study on the use of vitamin B17 against UV damage by using Drosophila as a model. 

-citations in the text as well as in the References must be given according to MDPI Authors' Instructions

Response: We appreciate the reviewer’s comment. We have carefully reviewed all citations in the text and the References section to ensure that they are formatted according to the MDPI Authors’ Instructions.

  • line 11 and others: all genus and species names must be written in italics, also in the References

Response: We thank the reviewer for this comment. All genus and species names have been checked and formatted in italics throughout the manuscript, including the References section.

  • line 65: authors should give reasons, why they used Drosophila as model organism

Response: we thank the reviewer for this comment. The rationale for using Drosophila melanogaster as a model organism has been provided in the Introduction lines 78-.80

  • line 113: there is no section 2.5 in the manuscript
  • Response: we thank the reviewer for this comment. This was a mistake and was deleted
  • line 138: centrifugation values must be presented in g, not in rpm
  • Response: we thank the reviewer for this comment. This value was presented as 11.180 g at line 160.
  • line 143: why Caspase here in uppercase letters?
  • Response: We thank the reviewer for this comment. The first letter of ‘Caspase’ is capitalized in accordance with standard scientific nomenclature for proteins and enzymes. ‘Caspase’ refers to a family of proteases involved in apoptosis.
  • line 146: v/v?
  • Response: We thank the reviewer for this comment. The abbreviation ‘v/v’ stands for volume/volume, indicating the proportion of one liquid in another.
  • line 147: explain acronym for horse reddish peroxidase here
  • Response: We thank the reviewer for this comment. The acronym ‘HRP’ has already been expanded as ‘Horseradish Peroxidase’ in line 172.
  • line 168: authors also used scanning EM, but did not mention this in Materials and Methods
  • Response: We thank the reviewer for this comment. A description of the Scanning Electron Microscopy (SEM) procedure has now been added to the Materials and Methods section lines 189-192.
  • legend to Fig.2: say that eggs were laid by 5-day old females within 24 hours
  • Response: We thank the reviewer for this comment. In the legend to Figure 2, it is already indicated that the egg-laying rate is expressed as the number of eggs laid per five females. This measurement does not indicate the age of the females, but reflects the total egg output over 24 hours as specified in the Methods section.
  • legend to figure 3: no statistics is shown in the figure 
  • Response: We appreciate the reviewer for this comment. Due to the layout and complexity of Figure 3, it was difficult to include letters indicating statistical group differences directly on the figure. Instead, the corresponding p-values are reported in the main text to convey the significance of the comparisons.
  • line 216: pupal
  • done
  • Fig. 6: data must be presented in columns. You cannot connect the results with line

       Response: We thank the reviewer for this comment. Figure 6 has been revised so that the data are now presented in columns, replacing the previous line connections.

  • line 250: data are shown in Table 2, not in Table 1. Table 1 can be deleted; the explanatiosn were given before.
  • Response: We thank the reviewer for this comment. Table 1 has been deleted, and the former Table 2 has been renumbered as Table 1 accordingly.
  • Table 2: better say, which are significantly different
  • Response: We thank the reviewer for this comment. The terminology in Table 2 has been revised to clearly indicate which values are significantly different.
  • Fig. 7: it would have been better using graphs with same image sections for better comparison
  • Response: We thank the reviewer for this comment. We carefully ensured that images were taken from the same region for all groups; however, the image shown in Figure 7 was selected as it most clearly illustrates the differences between the groups.

  • line 277: why uppercase letters here? Insert a space between number and unit (see also Fig. 8)
  • done
  • Fig. 10 C: use n or N, but not mixed
  • done
  • line 402: better say MDA level
  • done
  • References must be adapted to MDPI Authors' Instructions (see above).

Response: We thank the reviewer for this comment. All references in the text and in the references, section have been carefully checked and formatted according to MDPI Authors’ Instructions.

Round 2

Reviewer 1 Report

Comments and Suggestions for Authors

Although the authors did not fully improve the manuscript on my suggestions, they have give the proper reason, and improved the discussion section. I think it could be accepted after minor revised on the language.

Author Response

Although the authors did not fully improve the manuscript on my suggestions, they have give the proper reason, and improved the discussion section. I think it could be accepted after minor revised on the language.

Response: We thank the reviewer for the positive comment and for suggesting the manuscript for acceptance. A comprehensive language revision has been conducted, including grammar, clarity, and consistency checks. All identified issues have been corrected.

Reviewer 2 Report

Comments and Suggestions for Authors

Thank you for revising the manuscript and addressing most of my points. I find your justification for using a one-way ANOVA based on the experimental design to be well-reasoned. I have no further objections and am happy to recommend the manuscript for publication in its present form.

Author Response

Comment: Thank you for revising the manuscript and addressing most of my points. I find your justification for using a one-way ANOVA based on the experimental design to be well-reasoned. I have no further objections and am happy to recommend the manuscript for publication in its present form.

Response: We thank the reviewer for the positive comment and for suggesting the manuscript for acceptance. The requested information have been implemented in the revised manuscript.